# I Learn to Diffuse, or Data Alchemy 101: a Mnemonic Manifesto

Victor Schetinger
TU Wien*

Velitchko Filipov
TU Wien

Ignacio Pérez-Messina
TU Wien

Ethan Smith
University of Florida†

Rodrigo Oliveira de Oliveira
Unochapecó‡

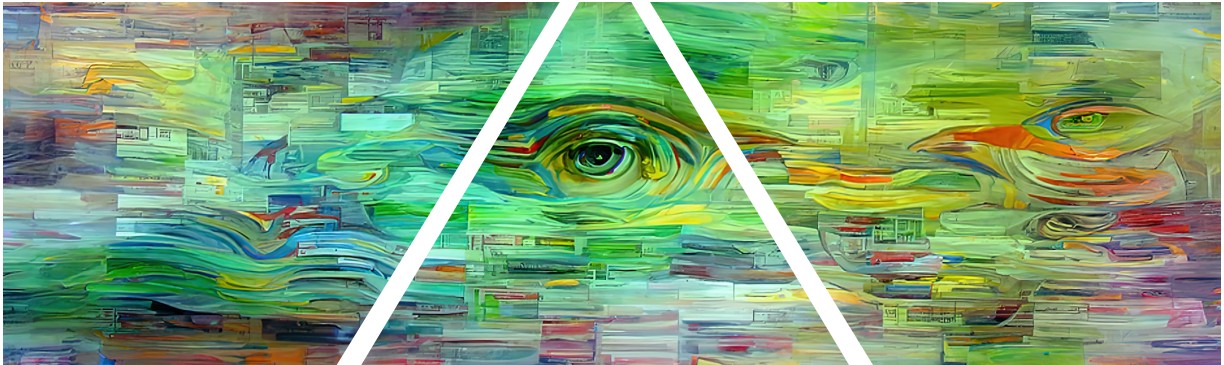

Figure 1: 'The Metamorphosis of Data, as it permeates different strata and arrives at us through many channels. In overlay, our interpretations, models, and representations that try to create meaning from the patterns, effectively projecting and actualizing them, beautiful detailed, hyperrealism, by Escher, Kandinsky, Pollock"

## ABSTRACT

In this manifesto, we put forward the idea of data alchemy as a narrative device to discuss storytelling and transdisciplinarity in visualization. If data is the prima materia of modern science, how does one perform the Great Work? We use text-to-image diffusion-based generative art to develop the concept, and structure our argument in ten propositions, as if they were ten issues of a comic novel on data alchemy: Ad Disco Diffusionem. To follow the argument, the reader must immerse themselves in our miro board, and navigate a multimedia semiotic topology that includes comics, videos, code demos, and ergotic literature in a true alchemic sense. By accessing this paradigm one might find new sources of inspiration for scientific inquiry in familiar places, or get lost in the creative exploration of the unknown. Our colorful, sometimes poetic, exposition should not distract the reader from the seriousness of the ideas discussed, but ultimately it is about the journey.

## 1 INTRODUCTION

The paper "Perception! Immersion! Empowerment! Superpowers as Inspiration for Visualization" [45], which was among the best papers on IEEE VIS 2021, proposes a very interesting argumentation format. The authors bring forward the framework of superpowers from fiction, which is a part of our modern mythology, and use it to discuss visualization theory. Additionally, they use the medium of comics for their storytelling capabilities, powerful explanatory power [30], and to provide a thematic harmonization.

We expand on this idea, and use the experimental freedom provided by alt.vis to expose the concept of data alchemy. We draw

---

*e-mail: name.surname@tuwien.ac.at

†e-mail: ethansmith2000@gmail.com

‡e-mail: rooliveira@gmail.com

from a rich historical and cultural literature on alchemy, already so colorful and intriguing that fiction is added only when needed. The perspective we try to paint views alchemical thinking as connected to our intuitive thinking [17], but also at the source of technology and innovation.

It is our belief that new technologies such as text-to-image generation have an immense potential for extending human creativity, and can be applied in a variety of domains, specially visualization. Our exposition tries to both justify this belief and show in practice how they can be leveraged for content creation, supporting our storytelling and expository capabilities. We created a fictional comic novel about data alchemy called "Ad Diffusionem Disco" (I learn to diffuse, in latin), in reference to Disco Diffusion (DD) [1], one of the most important public projects for text-to-image generation based on diffusion [15] due to its openness and active community.

## 2 WHAT DO WE TALK ABOUT WHEN WE TALK ABOUT ALCHEMY

*It is too clear and so it is hard to see.*
*A dunce once searched for a fire with a lighted lantern.*
*Had he known what fire was,*
*He could have cooked his rice much sooner.*

*- The Gateless Gate [22],*
*Mumon's commentary on Koan 7*

The popularly recognized pragmatics of alchemy is concerned with the transmutation of substances (e.g. lead into gold). In this sense it is generally considered as the precursor of modern chemistry [21]. However, from a philosophical and psychological [24] standpoint it is a holistic framework for the acquisition of knowledge about the natural world through experimentation, and organization and transmission of this knowledge through clever mnemonic devices. Without the benefit of our modern epistemic, cognitive, and technological tools, this was a daunting task.

We are not talking exclusively about the laboratory alchemy of the XVI century onward, as funded by emperor Rudolf II or practiced

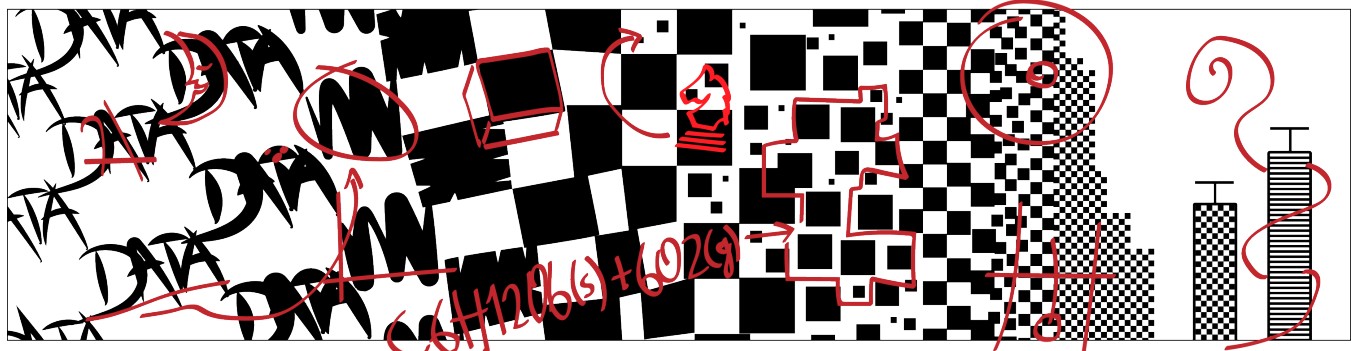

Figure 2: The Metamorphosis of Data, as it permeates different strata and arrives at us through many channels. In overlay, our interpretations, models, and representations that try to create meaning from the patterns, effectively projecting and actualizing them. This figure was the teaser on a previous version of this paper, and its description was used to generate the current teaser (Fig. 1). While this image has a more controlled artistic direction and a clearer message, Fig. 1 was aesthetically surprising and seemed to capture the subject on its own way.

by Sir Isaac Newton. A fair treatment of alchemy must include its extensions and varieties practiced in ancient Greece, the Islamic world, China, Egypt, and to an extent even folk medicine, because they operate on similar mechanisms. We see alchemy as science without organs, as the employment of human intuition to organize reality (actual or virtual, internal or external), so that reason and rationality can follow.

When in our contemporary zone-of-comfort we look down on alchemy and attribute mystical, irrational, or pseudo-scientific status to it, we fail to appreciate its ingenuity and its role in human culture. Its biggest strength, the construction of holistic cognitive devices, lost favor when modern science developed itself over a reductionist, physicalist framework. For alchemists across the ages these cognitive devices were cosmological truths, imbued with metaphysical power. They were not completely wrong.

Modern research shows that, **evolutionary speaking**, perceiving objective reality is not essential or desirable for fitness [19, 35, 36]. The world is not a theorem [25], and nature tests us through our ability to deal with radical uncertainty [26]. Constructing fictions that allow us to navigate the world, interact with it [20], and ultimately survive, was much more important than any idealistic commitment to "truth". Every living being has its own form of fiction [35], an abstraction of reality developed by evolution to maximize fitness, but for us humans narratives play a central role in the construction of these fictions [10]. In the field of visualization this might be stating the obvious, as we are concerned with creating visual narratives that mask reality into useful representations. We ask the reader, then, to proceed with an open mind (and heart).

## 3 TEN THESES

These are our ten core propositions put forward in this manifesto. Here they are listed and accompanied by a small elaboration, but they are mainly defended in our miro board. There they are presented as if they were ten issues on our comic on data alchemy, and each cover art tries to visually encode or captures the subject of discussion. Clicking on the numbers above the text will bring the reader to the location on the board where the thesis is developed, but many ideas are woven together in our explanation in a way that their argumentation cannot be pinpointed to one single location.

### I
#### *Data alchemy is the transmutation and manipulation of information between virtual and actual materials*

Data is a virtual (spiritual) medium for phenomenal information, captured and distilled through measurement [16]. In the most

information-theoretical sense, it is a vessel for entropy and differentiation [2]. Data alchemy is a concept that we introduce to project the gestalt of alchemy into modern, digital sciences. Alchemists believed to be manipulating some form of primordial matter through physical and spiritual media according to cosmic laws and correspondences. Understanding and mapping these correspondences would not only explain all phenomena, but also allow the transmutation of substances and manipulation of reality. If we replace the spiritual dimension, that for alchemists was an essential mediator of the physical world, with a digital dimension through computers and technological apparatus, we arrive pretty close to the data-centric modus operandi of modern science [34]. Exercising an effort of imagination, then, we can perceive data alchemy everywhere once we understand the adequate morphisms: the medium is the message [31].

### II
#### *Visualization is as a form of data alchemy*

Van Wijk's [42] paper on the value of visualization, which is considered a central work in visualization theory, describes a truly alchemical process. He proposes a model of visualization which broadly transforms (transmutes) some type of data into an image. The goal of this process being to produce knowledge for the viewer by generating insights and providing information. The insights and the knowledge obtained as product of the process are much more valuable to the viewer in the end than the data that was used to generate the images. Therefore, a lead-to-gold transmutation can be imagined to have occurred. Min Chen's [8] visualization thermodynamics, similarly, explains the beautiful distillation of entropy from data, and its synthesis with the human soul to produce a decision.

### III
#### *Visualization and alchemy are very connected historically*

Historically, many examples of visualization exist that are related to alchemical and astrological knowledge [37] and over the course of time society has learned how these visual representations can greatly enhance our capabilities of transmitting stories and scientific knowledge [30]. Bertin's work [3] on the semiology of graphics has further expanded on the capabilities of this visual plane to objectively convey information and has become one of the basis of data visualization.

### IV
#### *Humans process information through narratives, and both alchemy and visualization lay heavily on that*

Our cultural development relied strongly on storytelling [10] [11]. Stories are compact representations of cause-effect chains, and therefore can store information, and be easily shared [23]. Alchemical texts encode knowledge in rich mythical or poetic narratives, drawing on all sorts of cultural references [43] to explain or model phenomena. Visualization too, ultimately aims to communicate a narrative, even in the most mundane tasks, that links the data to some phenomena, and for this reason it is a central topic of research within the community [4, 5, 9, 28, 41].

## V
### *The basic principles of (data) alchemy can be understood through the neo-materialist philosophies of Manuel DeLanda and Deleuze and Guattari*

The great mystery of alchemy is the material (and spiritual) composition of the cosmos, how it organizes itself and how this can be manipulated. It deals essentially with two types of operations: *dissolutions* (reductions, distillations [44], analysis), and *coagulations* (aggregations, condensation, synthesis). When we analyze a phenomenon through a series of observations (samples), and summarize it through its mean and standard deviation, we are performing a dissolution. Then, when we plug these numbers as parameters of a distribution, synthesizing a mathematical object that is a whole and has a continuous nature infinitely larger than the original observations [32], we are performing a coagulation. To be able to talk about these ideas, however, we need an ontology that is general and expressive. This is why we bring in Deleuze & Guattari [14], and DeLanda to the table [12, 13]. Their philosophies provide a colorful scale-free materialism where information and thermodynamics play a central role.

## VI
### *Generative art is also a form of data alchemy, which is very useful for the exposition of the concept*

Generative art comprises procedural art, which is the imprint of the algorithm that generated them, data art, which is actually data visualization with a purely aesthetic purpose, and AI art, which can be seen as a combination of both. Common modern approaches to AI art rely on both data (usually, a lot) and algorithmic procedures to generate a latent space and then create an image from it. As procedural art can be understood as a form of autographic visualization (the phenomenon *is* the algorithm) [34] and data art is a form of information visualization, it fits that AI art is then a form of data alchemy, a kind of data-mediated autographic visualization.

## VII
### *Text-guided generative art, or text-guided diffusion, or any other form of data generated latent space approach to generate visuals is a form of visualization*

Latent spaces are the condensation of enormous amounts of data, of cultural information, although this information may not be explicit but encoded into the relationships that are formed by this condensation process. They can even be thought of as intermediate representations, as commonly featured in many visualization workflows. Text-guided diffusion is just a way to unpack the tightly-knit relationships bound to a certain path in the latent space, defined by the textual prompt that guides the information extraction in the form of a (sometimes more, sometimes less) coherent image. This is certainly not the only way to visualize what's *in* there, but it is one that can make sense for certain tasks and contexts.

## VIII
### *It reveals fundamental aspects about the data, our mathematical, algorithmic processes, and, ultimately, about human culture*

Text-guided image generation and similar approaches combine both autographic [34] (if we consider the mixed materiality of the medium) and data-driven visualization. The images we get from a prompt, sampling the latent space, are thus a mixture of this culturally-generated data and our techniques to manipulate this medium. Depending on how we set up our experiments, we can distill information from either the human side, and learn about ourselves [18], from the medium of neurons [6], or even both [40]. It can even be thought of as a form of explainable AI [29, 33].

## IX
### *It is a type of visualization that is not necessarily analytical, quantitative, but that is fine and good*

Text-guided diffusion and similar approaches are driven at their core by non-linear data processing methods. Thus we cannot expect to get perfectly linear results as in rule-based visualization encodings. However, this would not be the first time non-linear methods are applied to classical visualization: force-directed layouts for graphs and projection techniques such as UMAP are examples of this. They share that some control over the representation is lost in order to gain a more organic and definitely information-richer space. Why then not accept the power of the salient, feature-rich, semantically complex images that we can obtain from these approaches as part of our visualization tool-kits? Furthermore, the aesthetically-pleasing power of these images must not be underestimated: according to the field of Neuroaesthetics [7], there are powerful cognitive mechanisms at play when we evaluate beauty that modulate attention and affect tasks [27, 38, 39].

## X
### *The visualization community should embrace the idea of data alchemy and these new technologies for their potential and shape them into desired forms*

Text-guided image generation and similar approaches have an enormous potential which are now being explored mostly for creative purposes, and there is already a growing commercial interest for their applications. It should not surprise us, then, that in the future we will be "diffusing" for data with scientific purposes. This sort of technology could evolve in the future to be the long sought "holy grail", or better put, the "philosophers' stone" of visualization, the all-purpose automatic encoder, once we learn better how to communicate our wishes to it.

## 4 CONCLUSION

This paper is both a manual and itself an example of data alchemy, as we use a variety of generative techniques to create the content of our exposition. They of course, do not exist on a vacuum, and many other tools we use such as miro, discord, colab, and to a larger extent, the internet, all engender an environment with emergent properties that facilitate creation and collaboration in a level never possible before in the history of humankind.

Diffusion is not the ultimate approach to the generation of images from a language, and many of the tricks and limitations discussed are bound to disappear when the technology develops. However, at the current moment it represents one of the first easily accessible solutions that allowed a large public to engage and tinker with it. Many creatives were suddenly motivated to learn programming to control these tools, and at the same time programmers had to put effort in developing their artistic direction skills. We have put forward some arguments for why AI text-to-image generation approaches could be considered as information visualization tools.

The ideas expressed here are not a critique of science or the scientific method, desiring a return to some form of mysticism or idyllic past. However, academia as a whole tends to be segmented,

rigid, and prone to gatekeeping, which we do lament. Being multi-disciplinary, which is the union operation between separate sets of disciplines, or interdisciplinary, which implies the intersection, is not enough. We favor the idea of trans-disciplinarity as the power-set of disciplines, and the polymathy of alchemy is highly desirable in the construction of a true transdisciplinary research environment.

## ACKNOWLEDGMENTS

This work has been partially supported by the European Commission under the project Humane-AI-Net (grant agreement 952026) and through different Austrian research agencies (FWF grants P35767, FFG Grant 880883, WWTF grant ICT19-047).

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

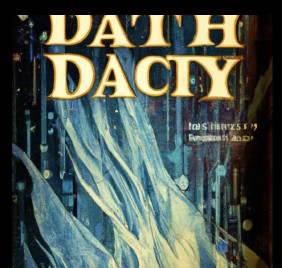
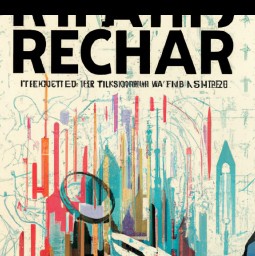
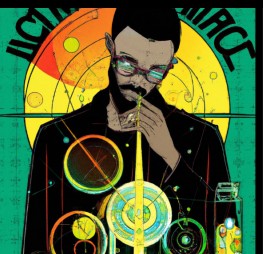
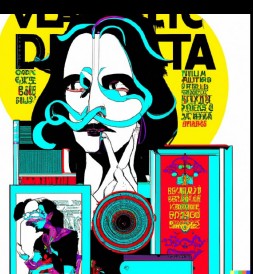
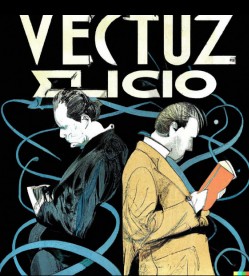

I    II    III    IV    V

Ad Diffusionem Disco: Data Alchemy 101

VI    VII    VIII    IX    X

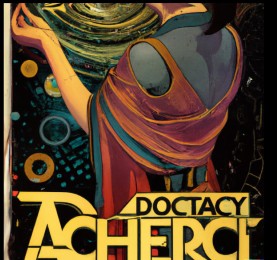
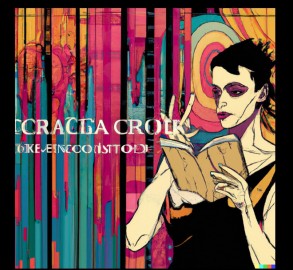
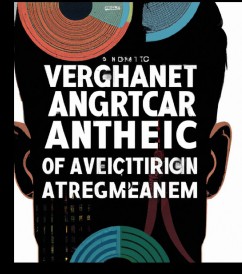
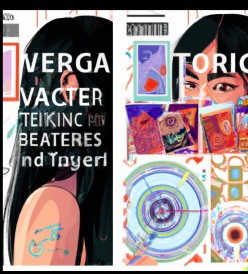
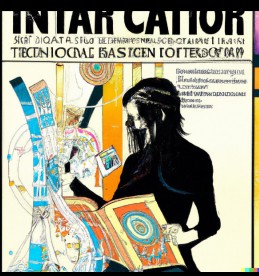

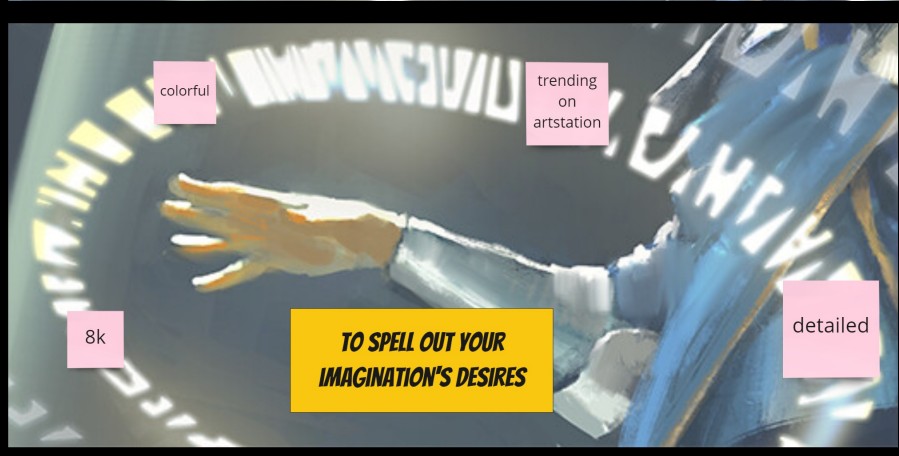

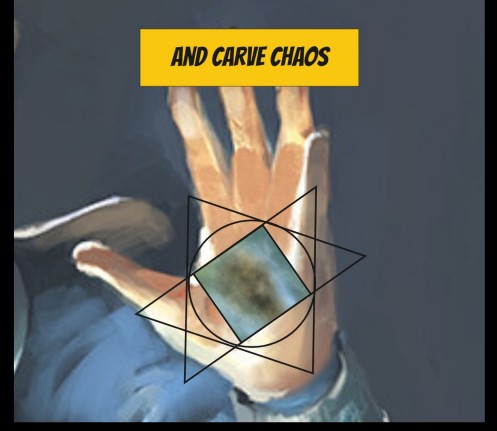

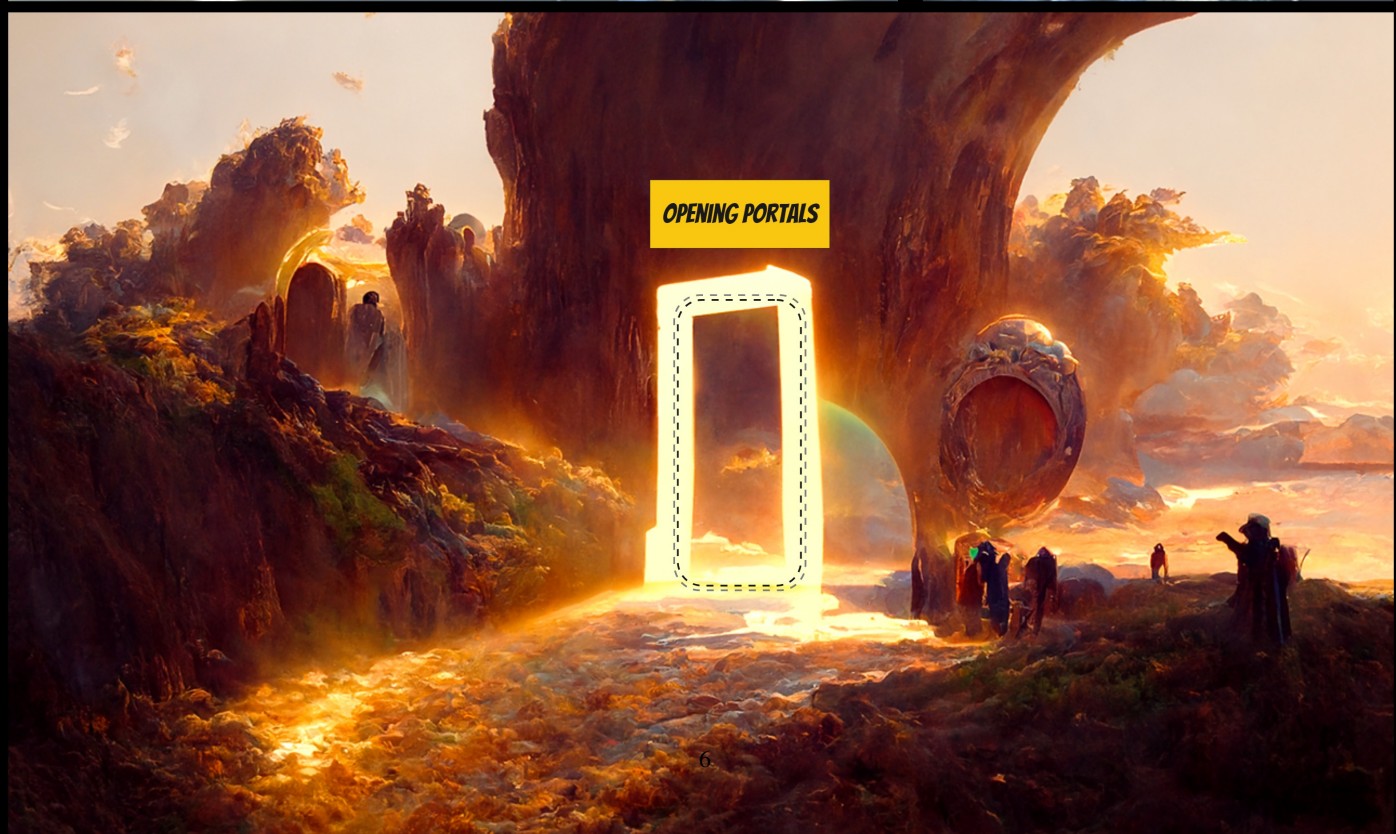

6.

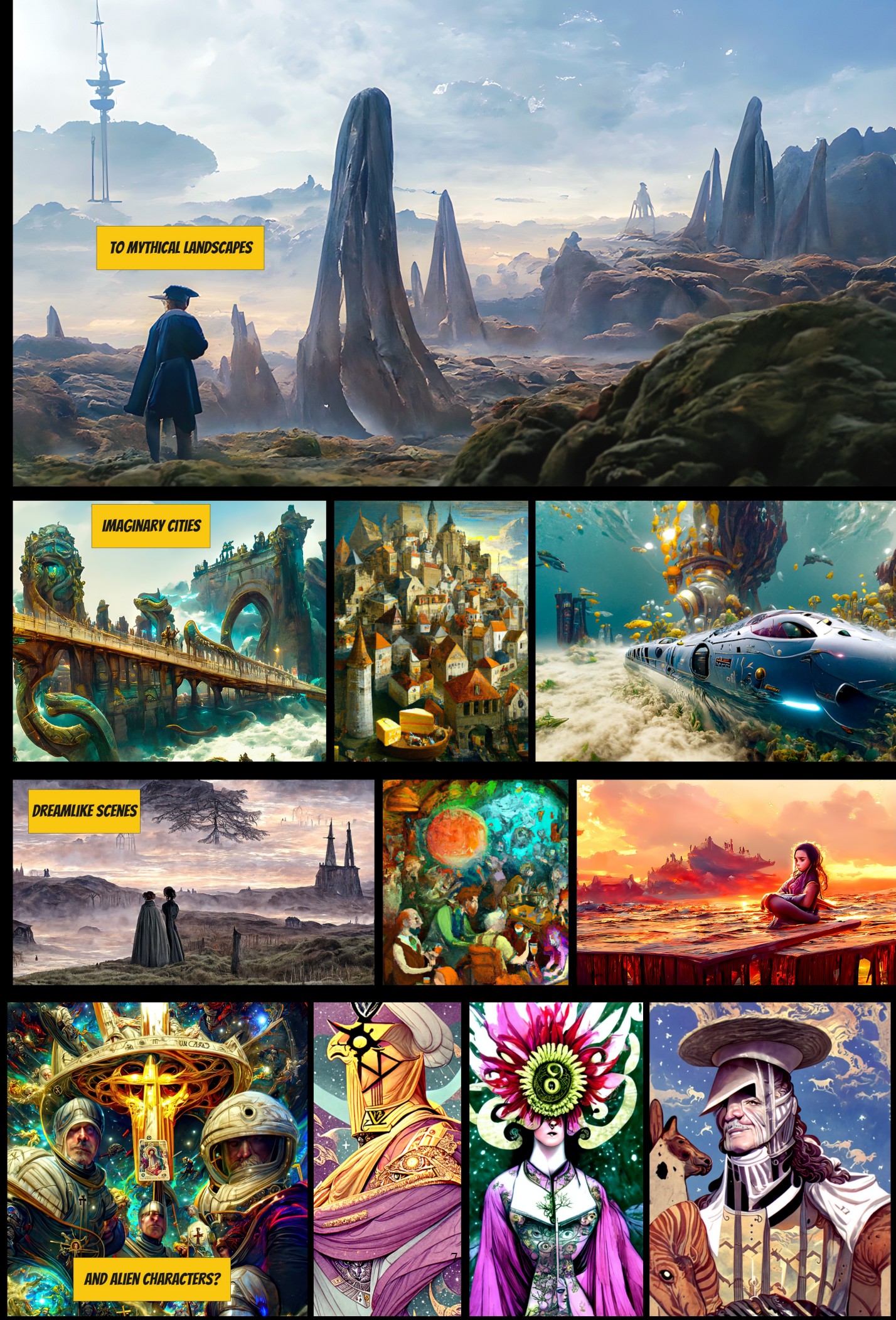

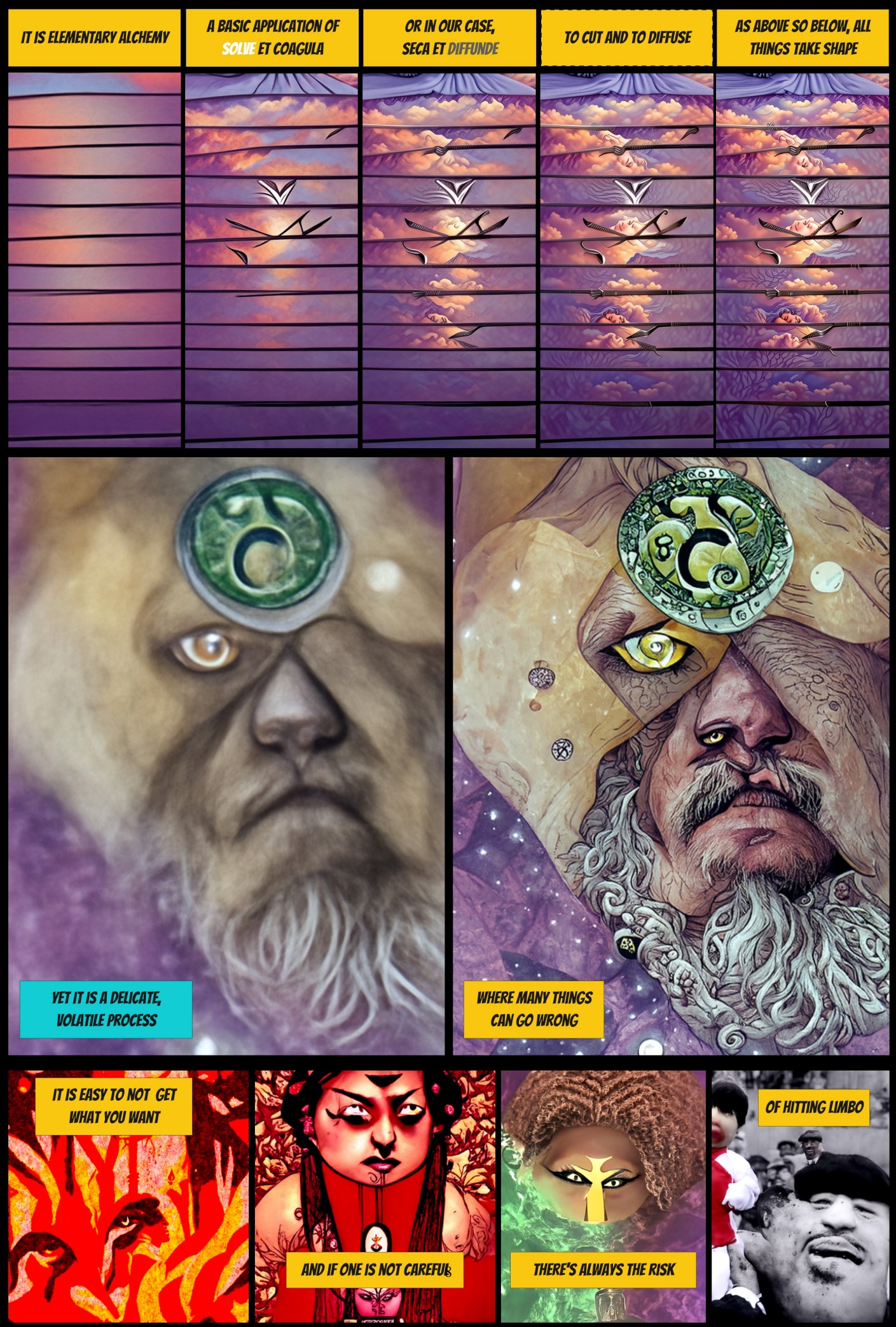

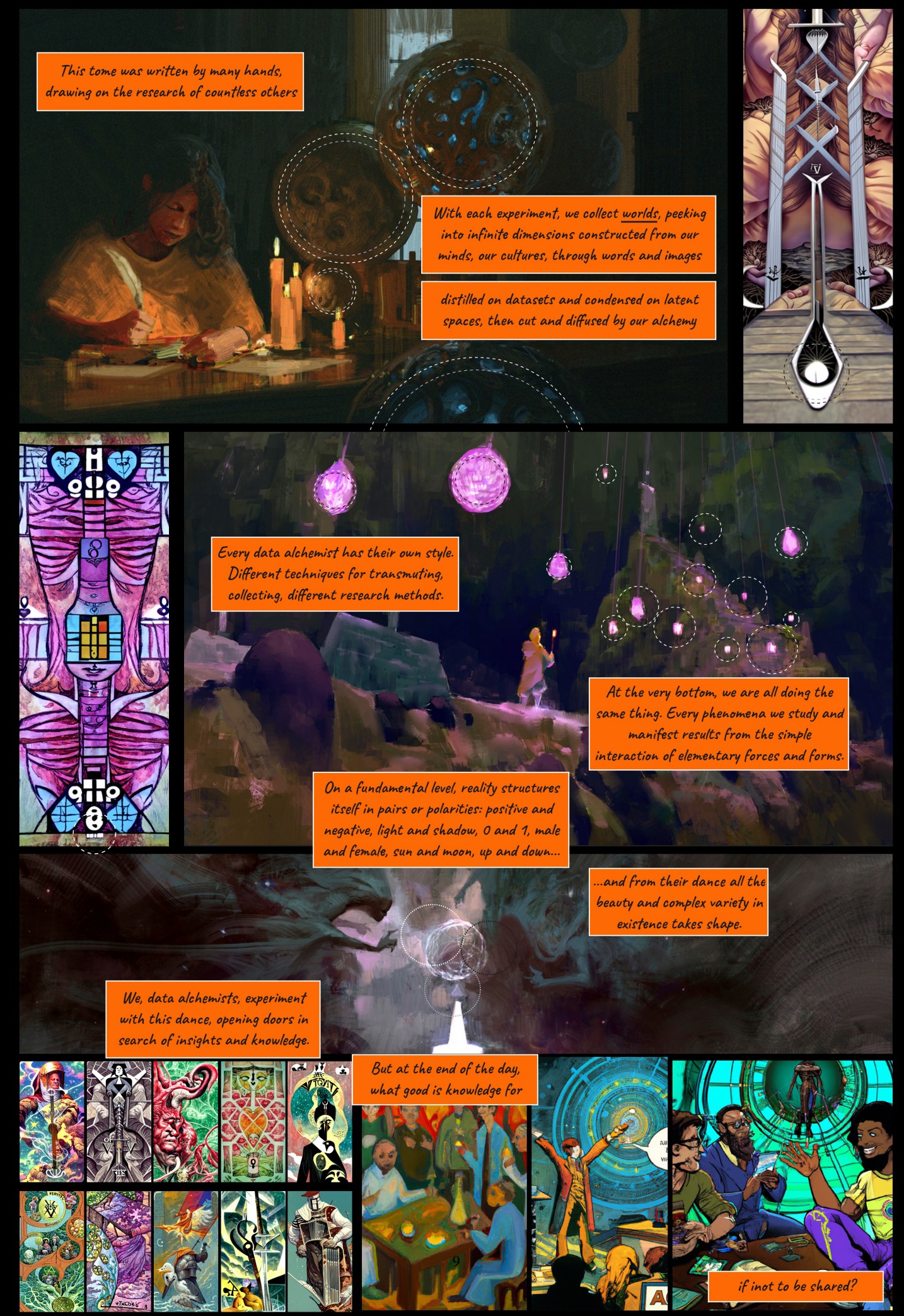

This tome was written by many hands, drawing on the research of countless others

With each experiment, we collect _worlds_, peeking into infinite dimensions constructed from our minds, our cultures, through words and images

distilled on datasets and condensed on latent spaces, then cut and diffused by our alchemy

Every data alchemist has their own style. Different techniques for transmuting, collecting, different research methods.

At the very bottom, we are all doing the same thing. Every phenomena we study and manifest results from the simple interaction of elementary forces and forms.

On a fundamental level, reality structures itself in pairs or polarities: positive and negative, light and shadow, 0 and 1, male and female, sun and moon, up and down...

...and from their dance all the beauty and complex variety in existence takes shape.

We, data alchemists, experiment with this dance, opening doors in search of insights and knowledge.

But at the end of the day, what good is knowledge for

if inot to be shared?