# OpenReview forum: "I Learn to Diffuse, or Data Alchemy 101: a Mnemonic Manifesto"
_IEEE.org/2022/Workshop/altVIS — Accept_

### Official Review · Reviewer_MZFU · 2022-08-10

**Review:**

This is beautiful work—a well-executed paper with a strong idea. I agree with the concept of generating images as a form of visualization of the latent space.

**Conflicts:**

None.

**Review Inclusion:**

Yes

**Sufficiently Alt:**

Yes

**Superlative:**

Most transformative.

---

### Official Review · Reviewer_KVuq · 2022-08-24

**Review:**

The research provides a comprehensive discussion of data visualization as alchemy.
Pros:
1. the paper is comprehensive in discussions that support data visualization as alchemy
2. the authors provide a supplementary micro board to further demonstrate the idea: https://miro.com/app/board/uXjVOmYzSgA=/?share_link_id=407698709002

Cons:
the authors should provide more discussions on the comparison between AI text-to-image generators to better demonstrate the connection and differences.

**Conflicts:**

NA

**Review Inclusion:**

No

**Sufficiently Alt:**

Yes

**Superlative:**

Perplexingist

---

### Official Review · Reviewer_zq6J · 2022-08-27

**Review:**

This paper proposes and explores the idea of data alchemy. It does so through a paper, parts of a comic book, and a miro board which variously seeks to highlight different components of this idea each of which contain various wonderfully rendered ml-generated images.

I think alchemy is a rich concept to draw upon for analogy within visualization and to motivate interesting work, however, I wish that it was more clearly defined. After having read the miro board and the paper I still don't entirely understand what data alchemy actually is. Roughly it appears to be something relating to the idea of interpreting text to image generation output as a data visualization (which is an idea I buy if you accept it as a visualization of algorithm and the training data, rather than something with an attributed agency, an issue that calling such entities AIs tends to give way to (although this paper avoids such a pitfall)), and then interpreting those results as a visualization that meant for enjoyment (which slots nicely into Munzner's enjoy/explore/present typification). Yet, I fail to see how that lens is different from the enjoyment of these images I had without applying the notion of data alchemy? I think I gather that it is supposed to be something learned from the provided koan rather than something provided directly, but I think it might be clearer if it was spelled out just a bit more explicitly.

Generally, I think this is compelling work that is beautifully presented. My central complaint is that, despite the volume of material, some of the central ideas are unclear. For instance, it would be useful if the abstract more clearly described the idea being proposed, and the consequences of doing so.

Some notes

- Brüggemann et al's work on applying Deleuze's notion of the fold to visualization seems deeply connected to thesis V
- It is asserted in thesis VIII that text-guided image generation can be seen as a form of explainable AI ``depending on how we set up our experiments'' which I do not believe is correct. It is creating an analogy that is not warranted (inference simply is not explainability) without justifying it. This perhaps could be argued more carefully if this is an important point to make.
- As is typical for me when I review papers, I printed the pdf, and find myself quite delighted by the physicality of the comic. I think it is rad and if this paper is accepted it would be wonderful to see a physical distribution of this comic (or an extended version of it!) at the workshop.
- I appreciated the history of alchemy, however, I wish that the same level of rigor and careful citation had been applied to its history as other parts. Similarly, It'd be cool if this work connected more clearly with prior work on vis + mysticism, or alt approaches to AI. These include my own vis/tarot work as well Browne and Swift's work on understanding how people understand AIs through seance and a variety of others.

**Conflicts:**

I do not believe I have any conflicts.

**Review Inclusion:**

Yes

**Sufficiently Alt:**

Yes

**Superlative:**

Most multi-modal

---

### Official Review · Reviewer_u8du · 2022-08-31

**Review:**

Meta-Review:

As your meta-alt reviewer I just want to start out by saying wow. What an excellent exploration of ideas that challenge the rigid positivism of data visualization’s onto-epistemology in a playful, yet rigorous way. It is my pleasure to say that the work is accepted.

I strongly recommend though that the paper include a clearer definition of alchemy and data alchemy (as suggested by Reviewer 1). Additionally, the paper would benefit from more clearly stating the intended consequences for accepting/following/engaging with your manifesto. I look forward to the presentation and encourage the authors to consider alt-y ways to engage the audience.


**Conflicts:**

None that I am aware of.

**Review Inclusion:**

Yes

**Sufficiently Alt:**

Yes

**Superlative:**

Most Likely to Concoct a Potion

---

### Decision · Program_Chairs · 2022-08-31

Accept